# Soil Abandonment as a Trigger for Changes in Zn Fractionation in Afforested Former Vineyard Acidic Soils

**Paula Pérez-Rodríguez** [1,2,*] **, Juan Carlos Nóvoa-Muñoz** [1,2] **, Manuel Arias-Estévez** [1,2]
**and David Fernández-Calviño** [1,2]

1   Departamento de Bioloxía Vexetal e Ciencia do Solo, Facultade de Ciencias, Universidade de Vigo,
    As Lagoas s/n, 32004 Ourense, Spain; edjuanca@uvigo.es (J.C.N.-M.); mastevez@uvigo.es (M.A.-E.);
    davidfc@uvigo.es (D.F.-C.)
2   Instituto de Agroecoloxía e Alimentación (IAA), Universidade de Vigo, Campus Auga, 32004 Ourense, Spain
*   Correspondence: paulaperezr@uvigo.es

**Abstract:** Zinc is an essential element for plant nutrition, but it may cause toxicity depending on its bioavailability and potential transformation in soil. In vineyard soils, high concentrations of Zn are usually found, mainly due to agricultural practices. However, a great abandonment of vineyards has recently occurred, leading to changes in the total and bioavailable Zn concentrations, as well as Zn fractionation. We analyzed Zn concentrations (total, $Zn_T$, and bioavailable, $Zn_{ED}$) and fractionation in the soil of three paired sites (PM, PT, and AR) up to depths of 50 cm in active and adjacent abandoned vineyards that were already transformed into forests. The $Zn_T$ averaged at 210 mg kg$^{-1}$ among all studied vineyards. The results showed changes in the vertical pattern $Zn_T$ concentrations after vineyard abandonment at the PM and PT sites, while at the AR site, no great variation occurred. The $Zn_{ED}$ (mean values = 7 mg kg$^{-1}$) decreased after abandonment at PM and AR in the uppermost surface layers, while it increased in the top 10 cm at the PT site, reaching up to 60 mg kg$^{-1}$. Regarding Zn fractionation in active vineyards, the residual fraction ($Zn_R$) was the most abundant, followed by Zn bound to crystalline Fe and Al oxy-hydroxides ($Zn_C$) and Zn bound to soil organic matter ($Zn_{OM}$). After abandonment, the $Zn_R$ slightly increased and the $Zn_C$ slightly decreased at the PM and AR sites at all depths, while the $Zn_{OM}$ showed a noticeable variation in the uppermost 10 cm of the PT site. These results suggest that the soil organic matter that is provided during afforestation may play an important role in Zn fractionation and mobilization, depending on its humification degree and chemical stability. Zn mobilization could result in a positive nutrient supply for plants, but caution must be taken, since an excess of Zn could cause toxicity in long-term abandoned vineyards.

**Keywords:** land-use change; afforestation; forest soils; zinc bioavailability; zinc mobilization; zinc–organic matter association

## 1. Introduction

Zinc (Zn) is an essential micronutrient for plants that plays a critical role in various physiological processes, including photosynthesis, hormone synthesis, and enzyme activation. Zinc occurs in natural soils at concentrations ranging from 40 to 120 mg kg$^{-1}$ depending on the soil parental material [1], but Zn concentrations are usually higher in agricultural soils due to the application of fertilizers, liming materials, manure, and, especially, fungicides containing Zn [2,3]. Zinc may accumulate in soils since it is easily adsorbed by minerals and organic compounds [1,4,5]. A zinc excess may result in toxicity for plants, leading to reduced growth and photosynthetic and respiratory rates, as well as imbalanced mineral nutrition and an enhanced generation of reactive oxygen species [6]. It also may affect soil organisms, leading them to develop tolerance when soil concentrations are higher than 125 mg kg$^{-1}$ in acid soils [7]. The availability and mobility of Zn in soil are influenced by its chemical forms or fractions, which can vary depending on soil properties and management practices [8–11]. A threshold of 200 mg kg$^{-1}$ of total Zn in soil was

established by the European Union (EU) to consider whether agricultural soil is safe or not for food production [12], while concentrations of available Zn that are higher than 15 mg kg$^{-1}$ are considered to be phytotoxic [1,13]. In vineyards, a Zn excess may decrease pigments and photosynthetic efficiency, and diminish leaves' activity, chlorosis, and even necrosis, among others [6].

The total Zn concentrations in vineyard soils from the NW of the Iberian Peninsula range from 100 to 170 mg kg$^{-1}$ in coastal areas, and from 60 to 149 mg kg$^{-1}$ in vineyards from inland areas, while the values for available Zn (Zn extracted with EDTA or DTPA) varied from 0.8 to 25 mg kg$^{-1}$ [13,14]. In other areas of the Iberian Peninsula, Zn concentrations in vineyards ranged between 16 and 154 mg kg$^{-1}$ [15]. At a global scale, total Zn was found in vineyard soils from Brazil, Slovenia, Greece, and Iran at high values ranging from 58 to 197 mg kg$^{-1}$ [16–19]. On the other hand, low concentrations of total Zn were found in vineyard soils of Brazil, showing severe deficiency, while low to adequate concentrations of available Zn were found in Italy, Brazil, or Syria [20–22].

The availability and mobility of Zn in soils are influenced by its chemical forms and/or fractions, which can vary depending on soil properties and management practices [8]. Organic matter, cation exchange capacity (CEC), and pH are known to play important roles in Zn retention on soil constituents by increasing its adsorption, and thus decreasing Zn mobility [8,23]. However, when Zn concentrations are in exceedance of the soil maximum adsorption capacity, Zn bonded with lower energy may be mobilized through the soil profile and water bodies [20]. In acidic soils, Zn tends to be strongly bound to soil components such as Fe and Al oxyhydroxides, limiting its uptake by plants [1,24]. However, land-use changes may modify the distribution of Zn in the soil's solid phase, influencing its availability and mobility in soil. The abandonment of agricultural soils, and the subsequent natural revegetation recovery, is a widespread phenomenon worldwide, but it is especially common in mountainous regions with steep slopes. Land abandonment occurred in Southern Europe at a proportion of around 25% up to 2011 [25], and about 11% of agricultural land in the EU is at a high potential risk of abandonment in the period of 2015–2030 [26]. It is expected that changes in the management of agricultural soil can affect its physicochemical properties and nutrient dynamics, influencing the availability and mobility of essential and non-essential elements such as Zn, Cu, Cd, and Pb. In the case of Cd and Pb, the time since vineyard abandonment was found to influence their bioavailable content, which increased as the time since the abandonment increased [27]. On the contrary, Cu concentrations were lower in an abandoned vineyard soil compared to an adjacent active vineyard, and even its fractionation changed towards a predominance of less mobile fractions [28]. Moreover, abandoned vineyards may experience changes in soil properties due to the cessation of management practices, potentially affecting Zn availability and speciation in soil [27,29].

Whilst several studies have determined the distribution of Zn in vineyard soils [5,16–18,30,31], none of these studies included abandoned vineyard soils. The effects that land-use changes can promote in the geochemical behavior of Zn deserve a thorough investigation in the case of abandoned vineyard soils, providing knowledge that could minimize potential negative environmental consequences derived from the cease of agricultural activities. Therefore, this study aimed to assess the distribution of Zn fractions in high-resolution sampled active vineyards (ACs) and their adjacent abandoned vineyards (ABs) in which a deciduous forest had grown, and how they changed because of the land-use changes. This will be analyzed from the perspective that Zn fractionation would be different between active and abandoned vineyards, this being a consequence of the changes in the soil properties and its management. The expected findings will provide insights into the effects of land-use changes on Zn cycling, assisting in future management strategies that promote the sustainable use of abandoned agricultural areas, and maintaining soil ecosystem services and biodiversity conservation.

## 2. Material and Methods

### 2.1. Study Area and Soil Sampling

Three sites were chosen for this study, named Portotide-O Mato (PM), Portotide (PT), and A Raña (AR). All these sites belong to the vine-growing area, the Ribeira Sacra Designation of Origin, which is partially spread along the valley of the river Miño (Galicia, NW Spain). Vine cultivation in this area has been practiced continuously since the late 19th century, and it is so-called "heroic viticulture" because it is developed in areas with very steep slopes (>30%) which have been skipped by arranging the land in terraces. Due to the hard work that the management of these vineyards demands, as well as the progressive aging of the local population, a great number of vineyards were abandoned, starting at least 35 years ago. Consequently, abandoned vineyards were progressively afforested with species that are characteristic of deciduous Atlantic forests (*Quercus robur* and *Quercus pyrenaica*), as well as some coniferous individuals of the *Pinus* genus (mostly *Pinus pinaster*). Nowadays, a huge number of old vineyards have become young Atlantic forests with the consequent land-use change.

The study area has a characteristic oceanic–Mediterranean transition climate, with 15 °C as the mean annual temperature and a rainfall depth ranging from 800 to 1300 mm year$^{-1}$. The local lithology is mostly dominated by schist (such as in the PT and PM sites) with small patches of granite and gneissic rocks (as in the AR site). The climatological conditions, also influenced by two water reservoirs located in the surroundings of the study area which provide a high degree of humidity during the vine-growing season, meant that the use of agrochemicals were widely applied for pest management.

The high-resolution soil sampling strategy was described in detail by Vázquez-Blanco et al. (2022) [28]. In brief, in each of the three sites, an active vineyard (AC) and an adjacent afforested abandoned vineyard (AB) were sampled up to a depth of 50 cm. Composite soil samples (made up of 5 subsamples) were obtained for each site and land-use type (active vs. abandoned) every 2 cm in the uppermost 10 cm of the mineral soil, every 5 cm between the 10 and 20 cm depths, and every 10 cm from 20 to 50 cm. This sampling strategy resulted in a total of 10 samples per soil type and land-use type. A composite soil sample, corresponding to the 0–20 cm depth range, was collected per site and land-use type to determine their general physicochemical characteristics. Before analyses, the soil samples were air-dried, sieved with 2 mm sized mesh, and homogenized.

### 2.2. Soil General Characterization

The general characterization of soil samples from the active and abandoned vineyard soils were previously shown by Vázquez-Blanco et al. (2022) [28] and included the pH in distilled water (pHw) and saline solution (pHk), the total contents of organic C and N, the exchangeable base cations (Na$^+$, K$^+$, Ca$^{+2}$, and Mg$^{+2}$) displaced with 1M NH$_4$Cl and exchangeable Al (with 1M KCl). The sum of exchangeable base cations and Al displaced with KCl was considered as an estimation of the effective cation exchange capacity (eCEC). In brief, active vineyard soils (ACs) are slightly acidic (pHw ranged from 5.6 to 6.0), with abandoned vineyards (ABs) being somewhat more acidic (pHw ranged from 4.6 to 5.3). The eCEC is quite similar in active and abandoned vineyards (range of 8.5 to 16.8 cmol$_+$ kg$^{-1}$), with Ca being the predominant exchangeable cation in all of them. The total organic carbon ranged from 15 to 24 g kg$^{-1}$ in active vineyards (ACs), and ranged from 36 to 71 g kg$^{-1}$ in the abandoned vineyards (ABs), whereas the ranges for the total N were 1.1–2 and 2.2–3.5 g kg$^{-1}$ for ACs and ABs, respectively. The total organic C (Z = −4.299, p = 0.000, N = 30) and N (Z = −2.251, p = 0.024, N = 30), as well as the C/N ratio (Z = −4.351, p = 0.000, N = 30), were significantly higher in abandoned than in active vineyards. The soil textures were sandy clay loam in the PM and PT sites, and sandy loam in the AR site. Detailed information on the soil chemical characteristics is provided as a Supplementary Material (Table S1).

*2.3. Total and Potentially Available Zn*

Total Zn ($Zn_T$) was extracted from the soil by digesting 0.5 g of soil with 5 mL of $HNO_3$, 4 mL of HF, and 1 mL of HCl in a microwave oven at 100 psi. For quality control purposes, two commercially available certified materials (i.e., soil, BCR 142R, and sediment, BCR 227R) were digested in triplicate following the same method. The Zn recoveries for both materials were consistent with the certified values (94.2 ± 2.8 and 183.2 ± 22.6 mg kg$^{-1}$, respectively). Potentially available Zn ($Zn_{ED}$) was extracted from the soil by shaking 5 g of soil in 50 mL of a solution containing 0.02M $Na_2$-EDTA and 0.5M $NH_4OAc$ at pH 4.65 for 1 h [32].

*2.4. Zinc Fractionation*

Zinc fractions were obtained following a non-sequential procedure [33]. Selective extractants were used to obtain operatively defined metal fractions in which Zn was bounded to different soil components, reporting their potential mobility in soil. So, Zn was measured in a soil extract derived from the application of 1M NH4Ac (pH 7) (Zn_ac), 0.1M Na-pyrophosphate (Zn_p), 0.2M oxalic acid–ammonium oxalate (pH 3) (Zn_o), and 0.2M oxalic acid–ammonium oxalate–ascorbic acid (pH 3.25) (Zn_ao). Consequently, the following operative fractions were obtained according to Zn bioavailability: (i) the exchangeable Zn fraction ($Zn_{EX}$), which is equivalent to that measured in Zn_ac extraction; (ii) the metal fraction bound to the soil organic matter ($Zn_{OM}$), resulting from Zn_p–Zn_ac; (iii) the metal bound to non-crystalline Fe and Al oxy-hydroxides ($Zn_{IA}$) obtained from Zno–Znp; (iv) the metal bound to crystalline Fe and Al oxy-hydroxides ($Zn_C$) after Zn_ao–Zn_o; and, finally, (v) the residual fraction of metal ($Zn_R$) obtained from $Zn_T$–Zn_ao. Zn bioavailability is expected to decrease in the sequence of $Zn_{EX} > Zn_{OM} > Zn_{IA} > Zn_C > Zn_R$.

All exchangeable cations (basic and acidic) as well as the Zn contents in the different extractions were determined via atomic emission spectrometry (Na and K) and atomic absorption spectrometry (Ca, Mg, Al, and Zn) using a Thermo Solaar M Series spectrophotometer (Thermo Fisher Scientific Inc., Franklin, MA, USA). Analyses of the total Zn and in the different extractions of Zn fractionation were performed in duplicate for 10% of samples, showing a variation coefficient below 5%.

*2.5. Statistical Analysis*

Non-parametric paired tests for 2 related samples were performed after significant results of Levene's test, meaning that the data obtained were non-normal and non-homogeneous in their variances. Tests using all data ($N = 60$) were performed to compare the results from the active and abandoned vineyard soils (factor: land use) with the soil characteristics, while tests per site ($N = 20$) were performed to compare the results on the influence of land-use changes for Zn fractionation parameters, since the total Zn levels were very different at each site.

Spearman's correlation analyses were used to test the correlation between the $Zn_{ED}$ and the total organic C (TOC) from the active and abandoned vineyard soils. All statistical analyses were conducted using IBM SPSS Statistics 25.

## 3. Results

*3.1. Total and Potentially Available Zn in Depth*

The total Zn ($Zn_T$) concentration in the 0–20 cm soil layer was >100 mg kg$^{-1}$ in all sites, independently of land use (Figure 1A,B). The zinc contents under both land-use types were higher in the PM and PT sites (ranging from 250 to 350 mg kg$^{-1}$) than in the AR site where the $Zn_T$ varied between 120 and 140 mg kg$^{-1}$. The total Zn concentration was slightly higher in the abandoned vineyards of the PM and PT sites, although differences regarding land use were significant only in the PM site ($Z = -1.988$, $p = 0.047$, $N = 20$). On the contrary, the $Zn_T$ levels in the AR site were slightly higher in the active than in the abandoned vineyard, but without a significant effect of land-use change. The highest $Zn_T$ concentration was found in the uppermost soil layer (0–2 cm) of the PT–AC vineyard, where it reached 353 mg kg$^{-1}$. In the active vineyards, the $Zn_T$ contents were generally

higher in the uppermost 10 cm, and then they gently decreased (as for PT–AC) or remained almost constant between 10 and 50 cm, as occurred in PM–AC and AR–AC (Figure 1A). In the first 10 cm of PT–AC, the $Zn_T$ showed two peaks at the soil layers of 0–2 and 8–10 cm. Regarding the abandoned vineyards, AR–AB showed a very narrow range of $Zn_T$ variation through the 50 cm of analyzed soil (118–130 mg kg$^{-1}$; Figure 1B), resulting in a vertical pattern that was quite similar to the corresponding active vineyard of that site. The variation in the $Zn_T$ with the soil depth in the PT–AB and PM–AB vineyards was somewhat different compared to their active ones. The PT–AB vineyard also showed two subsurface peaks for the $Zn_T$, but they occurred at different depths than in active vineyard (at 4–6 and 10–15 cm with 300 and 305 mg kg$^{-1}$, respectively). Below this secondary peak, the $Zn_T$ decreased progressively with the soil depth until reaching the lowest values of this site in the deepest soil layer (219 mg kg$^{-1}$). In the abandoned vineyard of the PM site, the lowest concentration of $Zn_T$ (217 mg kg$^{-1}$) was found at the surface soil layer (0–2 cm), contrary to that in the active vineyard. The vertical pattern of $Zn_T$ in PM–AB was more irregular than in PM–AC, with a double peak of $Zn_T$ in the subsurface layers (4–6 and 15–20 cm), but they did not coincide with those for PT–AB. Below 30 cm, the $Zn_T$ in PM–AB scarcely varied, and it was similar to those for PT–AB (240 mg kg$^{-1}$).

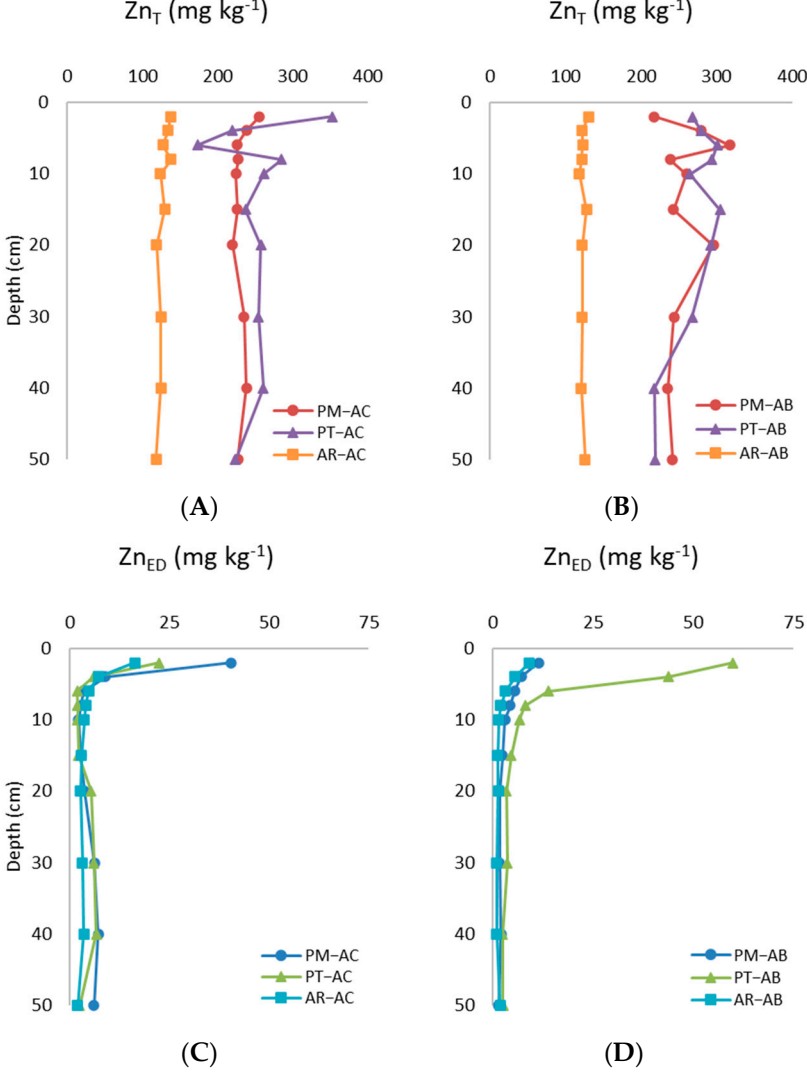

**Figure 1.** Distribution of total Zn ($Zn_T$) and potentially available Zn ($Zn_{ED}$) in active (**A,B**) and abandoned (**C,D**) vineyard plots.

The potentially bioavailable Zn ($Zn_{ED}$) in the active vineyards (Figure 1C) ranged from 2 to 40 mg kg$^{-1}$, following the sequence of PM–AC > PT–AC > AR–AC. It was noticeable that the site with the highest concentration of $Zn_T$ (PT–AC) was not the site with the highest concentration of $Zn_{ED}$. In the abandoned vineyards (Figure 1D), the $Zn_{ED}$ ranged from 0.6 to 60 mg kg$^{-1}$, following the order of PT–AB >> PM–AB $\approx$ AR–AB. The active vineyards showed greater values of $Zn_{ED}$ than the abandoned vineyards in the AR and PM sites, although the land-use change was a significant factor of variation for the $Zn_{ED}$ only for the AR site ($Z = -2.803$, $p = 0.005$, $N = 20$). The contrary occurred in the uppermost 15 cm of the soils from the PT site, where the $Zn_{ED}$ was higher in the abandoned than in the active vineyard. Regarding the vertical variation in the $Zn_{ED}$, all sites (including active and abandoned vineyards) showed almost the same pattern, and only minor differences were found. In the active vineyards (Figure 1C), the $Zn_{ED}$ was higher in all sites at the uppermost soil layers (0–2 cm), where it ranged between 16 and 40 mg kg$^{-1}$. Below 2 cm, the $Zn_{ED}$ decreased considerably up to a 6 cm depth, and then it remained relatively stable throughout the soil or even showed a slight increase between the 20 and 50 cm depths, but quite far from the values reached at the surface layers. Below the 6 cm depth, the $Zn_{ED}$ concentrations ranged between 0.9 and 8.0 mg kg$^{-1}$. In the abandoned vineyards, the highest values of $Zn_{ED}$ were also observed in the uppermost soil layer (0–2 cm), with values between 9 and 60 mg kg$^{-1}$ (Figure 1D). The values of $Zn_{ED}$ also decreased considerably in the upper 10 cm of the soil, but this was much less pronounced than in the active vineyards, except for PT–AB, whose $Zn_{ED}$ concentration decreased up to 10 times in the first 10 cm. As for the active vineyards, the levels of $Zn_{ED}$ in the abandoned vineyards showed few changes at depths below 20 cm (3.3 to 1.3 mg kg$^{-1}$), but they showed a trend at diminished progressively toward the deepest soil layer.

The percentage of available Zn with regard to the total Zn ($Zn_{ED}/Zn_T$ ratio) gives insights into the amount of potentially bioavailable Zn present in the studied soils. The values of the $Zn_{ED}/Zn_T$ ratio were generally higher in the active than in the abandoned vineyards in the PM and AR sites, particularly in the layers corresponding to the uppermost 6 cm. Below this depth, the proportion of available Zn regarding the total Zn was below 3%. On the contrary, in the PT site, the $Zn_{ED}/Zn_T$ ratio in the uppermost 6 cm of the soil was quite different between the abandoned and active vineyards, being 3–4 times higher in the former (Table 1).

**Table 1.** Comparison of $Zn_{ED}/Zn_T$ ratios (expressed as percentages) at the different studied sites.

| Depth (cm) | PM–AC | PM–AB | PT–AC | PT–AB | AR–AC | AR–AB |
|:---:|:---:|:---:|:---:|:---:|:---:|:---:|
| 0–2 | 16 | 5 | 6 | 22 | 12 | 7 |
| 2–4 | 4 | 3 | 3 | 16 | 5 | 5 |
| 4–6 | 2 | 2 | 1 | 5 | 4 | 3 |
| 6–8 | 1 | 2 | 1 | 3 | 3 | 2 |
| 8–10 | 1 | 1 | 1 | 2 | 3 | 1 |
| 10–15 | 1 | 1 | 1 | 1 | 2 | 1 |
| 15–20 | 2 | 1 | 2 | 1 | 2 | 1 |
| 20–30 | 3 | 1 | 2 | 1 | 2 | 1 |
| 30–40 | 3 | 1 | 3 | 1 | 3 | 1 |
| 40–50 | 3 | 1 | 1 | 1 | 2 | 1 |

Surnames –AC and –AB indicate active and abandoned vineyards, respectively.

### 3.2. Variations of Zn Fractionation with Soil Depth and Land Use

The variations in the different Zn fractions with the soil depth is shown in Figure 2. The expected mobility of Zn fractions, from the most to the least, is as follows: $Zn_{EX}$ > $Zn_{OM}$ > $Zn_{IA}$ > $Zn_C$ > $Zn_R$. In the active vineyards, the $Zn_{EX}$ (Figure 2A) ranged from 0.1 to 1.6 mg kg$^{-1}$, showing a very low variation with soil depth in AR–AC, while in the PM and PT sites, the $Zn_{EX}$ showed an irregular vertical pattern, with high values occurring equally in upper and deeper soil layers. In summary, the exchangeable fraction did not exceed 1% of total Zn in all active vineyards (Figure 3). In the abandoned vineyards, the

$Zn_{EX}$ (Figure 2F) ranged from 0.1 to 3.4 mg kg$^{-1}$, showing the same trend with soil depth as the active vineyards in the PM and AR sites. In PT–AB, the $Zn_{EX}$ was higher compared to the active vineyard, especially in the uppermost 20 cm, reaching up to 3.4 mg kg$^{-1}$ and showing a clear pattern of the $Zn_{EX}$ decreasing with depth (Figure 2F). In general, the $Zn_{EX}$ was higher in the active than in the abandoned vineyards of the PM and AR sites, although the differences according to the land-use changes were only significant in the AR site ($Z = -2.805$, $p = 0.005$, $N = 20$). The percentages of $Zn_{EX}$ regarding the total Zn were low in all former vineyards at any depth (Figure 3D–F), being only slightly above 1% of the total Zn in the uppermost 4 cm of the PT–AB soil (Figure 3E). As well as for the absolute values, the change in land use only brought about significantly different values for the percentage of $Zn_{EX}$ at the AR site ($Z = -2.871$, $p = 0.004$, $N = 20$).

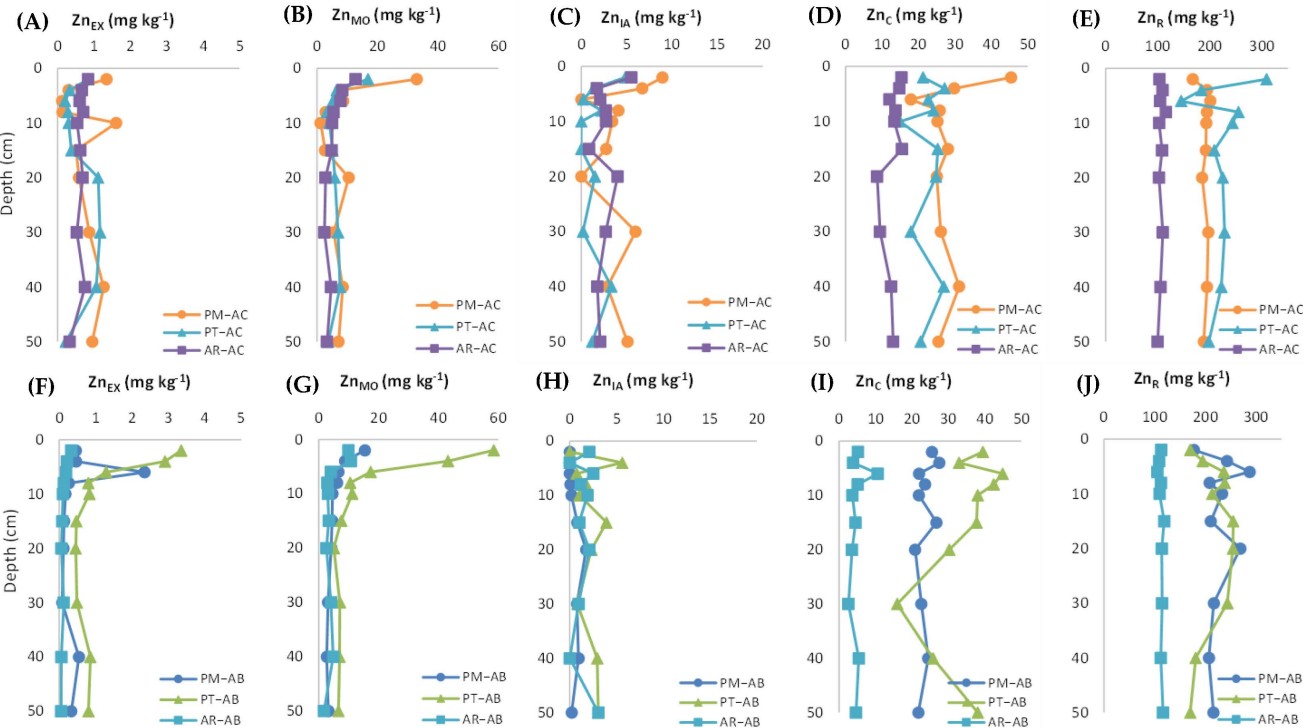

**Figure 2.** Distribution of exchangeable Zn ($Zn_{EX}$), Zn associated with soil organic matter ($Zn_{OM}$), Zn associated with inorganic amorphous oxides ($Zn_{IA}$), Zn associated with crystalline oxides ($Zn_C$), and residual Zn ($Zn_R$) with depth in active (**A–E**) and abandoned vineyard sites (**F–J**). Note that the X-axis has different graduations per fraction.

The Zn associated with the soil organic matter ($Zn_{OM}$) ranged from 1.2 to 33 mg kg$^{-1}$ in the active vineyards, and from 1.7 to 58.4 mg kg$^{-1}$ in the abandoned vineyards (Figure 2B,G, respectively). The land-use changes only significantly influenced the $Zn_{OM}$ values at the PT site, which increased after abandonment ($Z = -2.312$, $p = 0.021$, $N = 20$). Thus, the values of $Zn_{OM}$ in the upper 10 cm of the PT–AB vineyard were about 3 times higher than those showed in the active vineyard (Figure 2B,G). Contrarily, the $Zn_{OM}$ tended to be lower in the abandoned vineyards compared to the active ones in the PM and AR sites. Regarding its vertical pattern, the $Zn_{OM}$ showed a consistent trend in all sites under both land-use types, being higher in the uppermost surface layers (0–10 cm) and decreasing with the soil depth both in concentration and in percentage (Figure 2B,G; Figure 3A–F). As a percentage of the total Zn, the $Zn_{OM}$ fraction was especially remarkable in the topsoil sample (0–2 cm) of PT–AB, where it became the second largest fraction of Zn accounted for ≈22% of $Zn_T$ after vineyard abandonment and afforestation (Figure 3E). Except for the abovementioned layer and the 0–2 cm layer of the PM–AC soil, where the $Zn_{OM}$ achieves 13% of the total

Zn, the Zn fraction bound to the soil organic matter accounted for less than 10% of the total Zn in most of the soil samples (Figure 3).

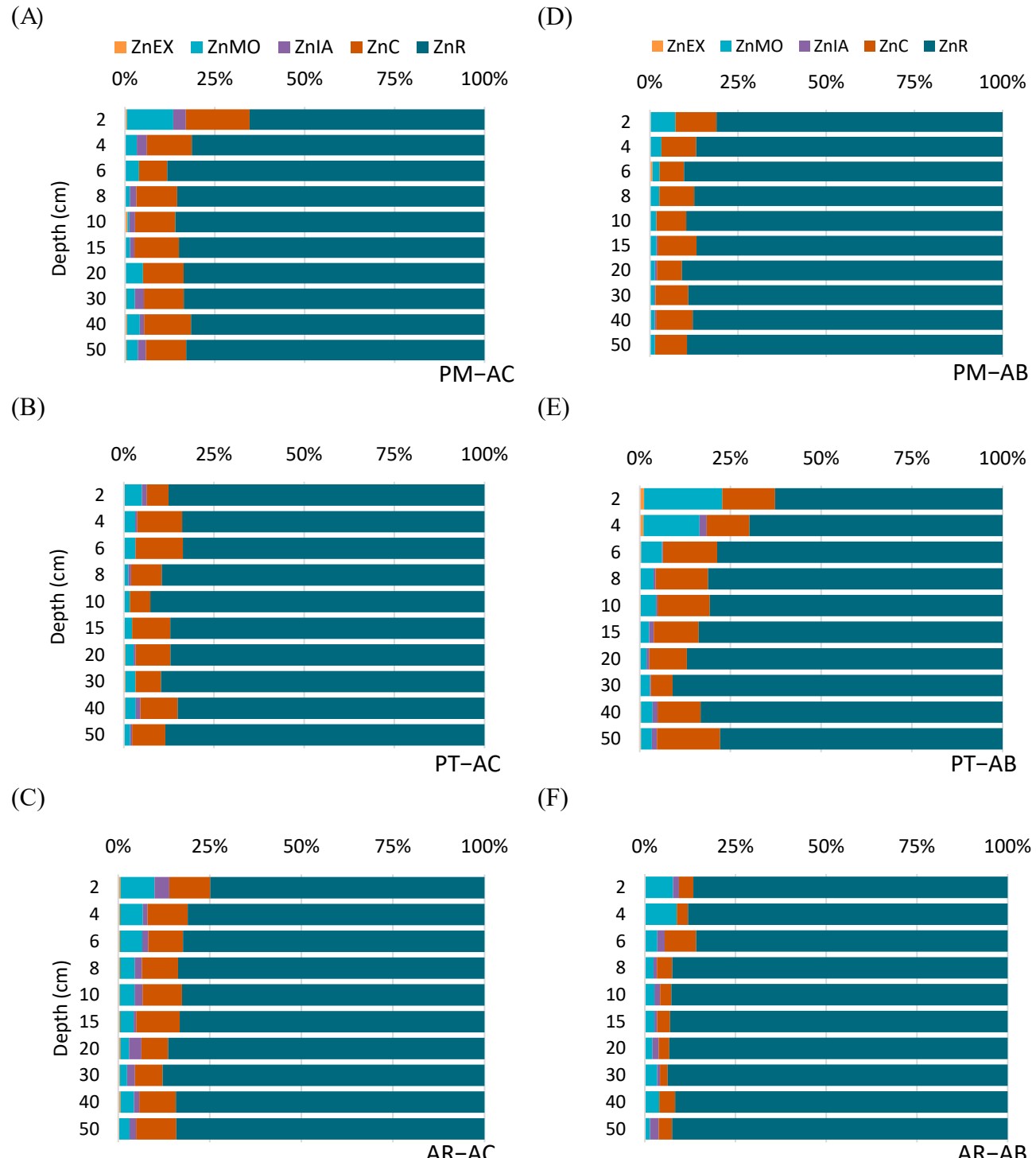

**Figure 3.** Percentages of exchangeable Zn (Zn$_{EX}$), Zn associated with soil organic matter (Zn$_{OM}$), Zn associated with inorganic amorphous oxides (Zn$_{IA}$), Zn associated with crystalline oxides (Zn$_C$), and residual Zn (Zn$_R$) per depth in active (**A–C**) and abandoned vineyard sites (**D–F**).

Regarding the Zn associated with inorganic amorphous oxides (Zn$_{IA}$), the pattern with the soil depth is quite irregular for the active vineyards in all sites, with values ranging from 0 to 9 mg kg$^{-1}$, and with the following sequence of abundance: PM–AC > AR–AC ≈

PT–AC (Figure 2C). However, after abandonment, the $Zn_{IA}$ generally decreased throughout the profile, with $Zn_{IA}$ values ranging between 0 and 5.6 mg kg$^{-1}$ (Figure 2H). Vineyard abandonment resulted in statistically significant differences of $Zn_{IA}$ in the PM ($Z = -2.547$, $p = 0.011$, $N = 20$) and AR sites ($Z = -2.091$, $p = 0.037$, $N = 20$), with greater values in the active vineyards (Figure 2C,H). The levels of $Zn_{IA}$ were always lower than those of the $Zn_{OM}$, being almost 4 and 10 times lesser in the active and abandoned vineyards, respectively. As a percentage of the total Zn, the land-use change resulted in a decrease in the fraction of Zn associated with inorganic amorphous oxides, with values below 2% in the abandoned vineyards and at about 4% in the active ones (Figure 3A–F). The percentages of $Zn_{IA}$ regarding the total Zn were significantly lower in the abandoned than in the active vineyards in the PM ($Z = -2.547$, $p = 0.011$, $N = 20$) and AR ($Z = -2.143$, $p = 0.032$, $N = 20$) sites, a fact that was particularly reflected in the superficial layers (0–4 cm).

In general, the Zn associated with crystalline Al and Fe oxyhydroxides ($Zn_C$) is the second fraction of Zn in order of abundance, ranging from 3 to 46 mg kg$^{-1}$ in all analyzed samples (Figure 2D,I), except for the uppermost samples (0–4 cm) of the PT–AB vineyard, where this fraction is surpassed by the $Zn_{OM}$ (Figure 2B). As regards its vertical pattern, the $Zn_C$ also showed an irregular trend with the soil depth. For instance, a maximum value (45 mg kg$^{-1}$) was registered at the uppermost soil layer (0–2 cm) of the PM–AC soil, whilst in the rest of the profile, the $Zn_C$ values remained at about 25 mg kg$^{-1}$ (Figure 2D), whereas in the PT–AC and AR–AC vineyards, the $Zn_C$ showed a saw-toothed pattern ranging from 10 to 15 mg kg$^{-1}$ (Figure 2D). The abandonment of vineyards showed two different behaviors depending on the site. In the PT vineyard, the $Zn_C$ concentrations were significantly higher in the abandoned vineyard compared to the active one ($Z = -2.397$, $p = 0.017$, $N = 20$). Contrarily, the active vineyards showed greater values of $Zn_C$ than the abandoned vineyards in the PM ($Z = -2.803$, $p = 0.005$, $N = 20$) and AR ($Z = -2.805$, $p = 0.005$, $N = 20$) sites (Figure 2D,I). As a percentage of the total Zn, the $Zn_C$ fraction ranged from 2 to 18%, depending on the site and soil depth (Figure 3A–F), being significantly affected by the land-use changes in the three sites ($Z = -2.397$, $p = 0.017$, $N = 20$ in PT; $Z = -2.547$, $p = 0.011$, $N = 20$ in PM; and $Z = -2.143$, $p = 0.032$, $N = 20$ in AR).

The residual Zn ($Zn_R$), i.e., the least mobile fraction, ranged from 100 to 309 mg kg$^{-1}$ and showed quite a homogeneous vertical pattern in the active vineyard soils in PM–AC and AR–AC, whereas a considerable diminution in the $Zn_R$ values occurred in the PT–AC soil between 2 and 6 cm (Figure 2E). The lowest values of $Zn_R$ were found in the AR–AC soil, with values of around 100 mg kg$^{-1}$. In the abandoned vineyards, the values of $Zn_R$ as well as its vertical pattern in AR–AB were similar to its equivalent active vineyard. In the other two abandoned vineyards (PT–AB and PM–AB), the $Zn_R$ showed an even more irregular vertical pattern than their corresponding active vineyards. Thus, the $Zn_R$ tended to increase in the uppermost 10 cm in both soils, with some secondary peaks at 15 and 20 cm depth, and finally, a progressive decline towards the deepest soil layers (Figure 2J). The $Zn_R$ fraction represented more than 75% of the $Zn_T$ in the soil of all vineyards, independently of the land-use type (Figure 3A–F). Only some of the uppermost soil layers (0–2 and 2–4 cm) depart from this general trend, as occurred for PM–AC (Figure 3A) and PT–AB (Figure 3E), where the percentage of $Zn_R$ ranged between 50 and 75% of the total Zn.

## 4. Discussion

### 4.1. Total and Available Zn Concentrations

The total Zn concentrations found in the abandoned and active vineyard soils of the PM and PT sites were similar to those reported for the active vineyard soils in the same study area by Campillo-Cora et al. (2019) [34]. Our values were somewhat higher than those reported by Fernández-Calviño et al. (2012) [14] and Wightwick et al. (2008) [35] for different wine-growing areas in NW Spain (60–150 mg kg$^{-1}$) and Australia (14–161 mg kg$^{-1}$), respectively. On the other hand, the values of total Zn in the AR site were in the range of the mentioned studies. However, all the values of $Zn_T$ that are reported in the present study were several times higher than the mean value (41 mg kg$^{-1}$) that was recently reported by

Van Eynde et al. (2023) [36] in topsoil across Europe, suggesting the occurrence of an excess of Zn.

The difference in the total Zn between the PM and PT sites compared to the AR site may be due, to some extent, to a distinctive contribution of parent materials, as their characteristics may determine Zn levels in topsoil [37]. Thus, the average geochemical background level of Zn in soils developed from granite in NW Spain was stablished to be 51 mg kg$^{-1}$, slightly below the level that is considered for schist soils, which is 68 mg kg$^{-1}$ [38]. Therefore, the distinct parent material could partially explain the differences observed in the total Zn among the PM and PT sites regarding the AR site. Despite this, it is necessary to assume that a Zn excess above the geochemical background values in all sites results as a consequence of the agricultural activities that were carried out during vineyard management, and that those should be much more intense in the PT and PM sites than in the AR site. Soil pollution with Zn was reported as consequence of agricultural practices related to fungicide application [39], over-fertilization [3,14,40,41], the addition of organic amendments, and the supply of biosolids containing Zn [42,43]. As the abandoned vineyards were also affected by different agricultural practices before afforestation, they can be also considered as soils with a certain Zn pollution degree. Although the historical application of Zn-based fungicides and fertilizers in the studied soils is unknown, these moderate-to-high values of total Zn could be due to the use of mancozeb and Zn phosphates in the study area.

The occurrence of the highest total Zn in the uppermost centimeters of the active vineyards is consistent with a superficial supply of fertilizers and amendments. The studied active vineyards are composed of narrow terraces (sometimes with space for only one vine row) arranged in a steep slope area, which preclude the use of machinery, and the management practices are restricted to shallow tilling by hand ("heroic viticulture"). Moreover, organic matter is mainly accumulated in the few uppermost centimeters of the vineyard soils [28], and its strong affinity to heavy metals in the surface soil layers of active vineyards creates a favorable geochemical environment for Zn enrichment. Something similar occurred in abandoned plots where, in addition to the legacy of their past management as vineyards, Zn could also reach the uppermost soil layer through litterfall, as it is an essential micronutrient for plants [44]. The additional contribution of Zn from decaying aboveground biomass would explain the higher values of total Zn found in some of the abandoned vineyards than in the active ones (as in the PM and PT sites). This accumulation of Zn in the soil surfaces of abandoned vineyards is consistent with the role of the uppermost organic-matter-rich layers of forest soils as a temporal sink for Zn [45]. The trend in Zn accumulation in the uppermost soil layers observed in the present study agrees with previous research on vineyard soils worldwide [20,46].

The general decreasing trend of the total Zn with the soil depth is expected, considering the strong association of Zn and organic C, and the vertical variation of the latter. Despite this, the observed peaks in the total Zn at subsurface levels in the PT and PM sites can be attributed to the different soil depths that were reached with the agricultural practices, as well as a natural variability in the lithogenic Zn levels. However, this does not preclude that Zn could be mobilized to deeper soil layers, bound to low-molecular-weight organic acids [47,48]. This potential mobility of Zn along the soil profile could help to explain the homogeneous vertical pattern of the total Zn in the vineyards of the AR site, a fact that could be facilitated via the sandy texture of these soils, derived from granitic rocks.

Regarding the potentially bioavailable Zn (Zn$_{ED}$), the range of values in the present study is wider than that (0.8–11 mg kg$^{-1}$) reported for vineyards soils in NW Spain by Fernández-Calviño et al. (2012) [14]. Our values are also slightly higher than those reported by Brunetto et al. (2018) [40] in Brazilian vineyard soils, which reached around 40 mg kg$^{-1}$ in the uppermost soil layers, or those recently published for vineyard soils in wine-growing areas of Hungary, which ranged from 1 to 6 mg kg$^{-1}$ [49]. It is important to note that the highest Zn$_{ED}$ concentrations tended to be found where the values of TOC were higher, particularly in the uppermost soil layers of the active and abandoned vineyards (Figure 4). In fact, the TOC and Zn$_{ED}$ were significantly correlated when all samples were considered

(rho = 0.276; P = 0.033, n = 60). This relationship is expected because organic matter has a strong affinity for metals and provides available sites for Zn binding in active vineyards, whereas in abandoned vineyards, the accumulation of organic matter derived from litterfall in the uppermost soil layers would also contribute to Zn complexation. When the vertical pattern of the $Zn_{ED}$ is examined in the active and abandoned vineyard soils, the C/N ratio is the parameter that showed a better correlation (rho = 0.485; P = 0.000, $N$ = 60). This suggests that, rather than the quantity of total organic matter, the $Zn_{ED}$ distribution in the active and abandoned vineyards is the most dependent on the humification degree of the soil organic matter. So, despite the total Zn concentration being lower in the abandoned vineyards, it may result in more bioavailability than the active vineyards if the Zn is complexed by organic matter, resulting in Zn mobilization being dependent on soil organic matter (SOM) mineralization. Mineralization is mainly dependent on edaphoclimatic factors, such as temperature, humidity, the abundance of decomposers, and the type of organic matter formed [50]. Considering that fresh litters (in abandoned vineyards) usually contain little N compared to amount that the decomposers need (C/N ratios ≥ 20), the mineralization process is slow [51,52] and, therefore, eventually less concerning.

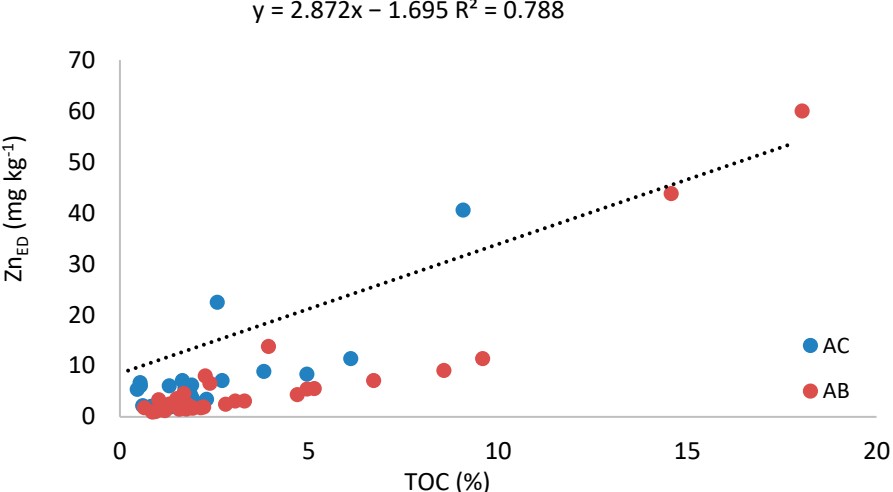

**Figure 4.** Relationship between the concentration of available Zn ($Zn_{ED}$) and TOC in active (blue dots) and abandoned (red dots) vineyards.

Additionally, our values of $Zn_{ED}/Zn_T$, even those of the abandoned vineyards, were lower than those obtained by Brunetto et al. (2018) and Brunetto et al. (2014) [17,40] in Brazilian acid vineyard soils, which ranged from 30 to 52% in the uppermost soil layers. However, our results were higher than those found in neutral to alkaline vineyard soils from the Tokaj region (Hungary) and from alkaline vineyards from NE Spain, which obtained ratios below 7 and 10% in the upper 20 cm [49,53]. Such values mean that a small percentage of the total Zn may be obtained by plants without inducing toxicity, and that the soil's pH is a key parameter in driving Zn bioavailability.

Nevertheless, values of $Zn_{ED}$ deserve special attention and must be examined, especially in the uppermost soil layers of most of the vineyard soils (active and abandoned) that have been studied, since, depending on concentrations, they can exceed the phytotoxic threshold for agricultural crops [1] and may result in environmental concern. Thus, caution must be taken when new agricultural crops might be planted into those sites, as although high concentrations of $Zn_{ED}$ have been previously found in active vineyards exceeding the limits of phytotoxicity [13], they have also found in long-term abandoned vineyards. However, the relatively high levels of $Zn_{ED}$ found in the studied soil could favor the natural instauration of native trees after abandonment, as trees usually require a good supply of Zn for optimal nutrition [54].

### 4.2. Changes in Zn Fractionation after Abandonment

Generally, >75% of Zn belongs to the residual fraction at all studied depths, except for PT–AB up to the 4 cm depth. This was also found by Beygi and Jalali (2019) [16] in vineyard soils from Iran. Similar results were found in vineyards from France [9], Brazil [5], and from other designations of origin in NW Spain [55]. Zahedifar (2017) and Korchagin et al. (2020) [5,56] found a predominance of the Zn residual fraction in soils under different land uses, including agricultural (cereals and vineyards, respectively) and forest soils, which indicates that, generally, Zn predominates as being geochemically non-mobile, independently of land use, and indicating that its origin is mainly natural. In the present study, the more abundant Zn fractions followed the sequence of $Zn_R > Zn_C > Zn_{OM}$. Generally, a level of $\geq$90% corresponded to the least mobile fractions, i.e., Zn bound to Fe and Al non-crystalline ($Zn_{IA}$), crystalline oxy-hydroxides ($Zn_C$), and the residual fraction ($Zn_R$). The exceptions were found in PM–AC at the upper 2 cm, where the Zn mobile fraction ($Zn_{EX} + Zn_{OM}$) reached 13%, and in the uppermost layers (0–2 and 2–4 cm) of PT–AB, where the Zn associated with mobile fractions reached 23 and 16%, respectively. The higher concentration of SOM in the uppermost soil layers may explain the differences in Zn fractionation in the surface layers, since SOM can mobilize Zn through the complexation or chelation processes via organic functional groups, improving the humic substances–metal interactions [5].

In the active vineyards of the PM and AR sites, two out of the three most abundant fractions ($Zn_R$ and $Zn_C$) were barely constant with depth, especially below 10 cm, suggesting that this homogeneity could be consistent with lithology as the main Zn source in these sites and, therefore, showing a lesser influence of the land-use type. Our results hardly differ from those obtained by Korchagin et al. (2020) [5], in which they found a smooth decrease in $Zn_R$ with a pronounced decrease in Zn linked to minerals. However, the Zn fractionation changed after abandonment, since the $Zn_C$ decreased, while the $Zn_R$ slightly increased. This could be explained by the ageing of $Zn_C$ forms and their progressive transformation to more recalcitrant Zn forms that become part of $Zn_R$, a process that may be more favorable to occur when no Zn addition due to anthropogenic activities takes place [57]. At these sites, the $Zn_{OM}$, which resulted in being the third most abundant fraction, varied with depth, tending to be higher at the soil's uppermost 4 cm, possibly due to a greater occurrence of organic matter supply from aboveground tree biomass deposition. Similar results were also found by Korchagin et al. (2020) [5] regarding the abundance of this fraction and its importance on surface layers, both in vineyard and forest soils. Despite the Zn bounded to SOM showing a trend to slightly decrease with soil depth, it did not vary significantly after the land-use changes (Figure 3A,C,D,F). At the PM and AR sites, such a diminishing trend in the $Zn_{OM}$ could be due to a less abundant litterfall input or its occurrence during a shorter period, resulting in a modification of this Zn fraction in only the uppermost cm of soil (Figure 3D,F).

The PT site showed a different behavior regarding Zn fractionation. Firstly, the OM fraction increased after abandonment, being quite different between the active and abandoned vineyards at the top 10 cm as a result of the continuous contribution of the senescent biomass of the vegetation growth with the progress of the abandonment. Secondly, the Zn bounded to OM was the second most abundant fraction at the top 4 cm (Figure 3E), which could be expected considering the increase in organic matter and its affinity to metals. The proportion of zinc bounded to OM increased from 5 and 3% (Figure 3B) to 22 and 15% at 2 and 4 cm depths (Figure 3E), respectively. It should be noted that the $Zn_{EX}$ also increased from values near to 0 (Figure 3B) to being higher than 1% (Figure 3E). Consequently, the most mobile Zn fractions ($Zn_{OM} + Zn_{EX}$) reached nearly 25% of the total Zn after abandonment, while in the active vineyards, these values were lower than 5%. The previous hypothesis of a longer period of abandonment at the PT site could explain the increase in the most mobile fractions after transformation due to a probable higher degree of SOM humification. These results must be considered, since Khoshgoftarmanesh et al. (2018) [58] found a positive correlation of $Zn_{EX}$ and $Zn_{OM}$ with the Zn uptake in the above-

ground biomass of wheat cultivars, meaning in the present study that high concentrations of SOM may mobilize Zn to be used by trees or wild plants that are naturally grown in the new afforested sites. Additionally, the $Zn_C$ generally increased after abandonment at all depths, resulting in an overall diminution of $Zn_R$. Consequently, Zn fractionation becomes modified (especially in the topsoil) after abandonment in favor of the most mobile fractions, increasing its geochemical mobility in the uppermost soil layers. This could enhance the Zn toxicity risk for plants and soil microorganisms, although this concerning scenario would also depend on the SOM composition as well as its humification degree. The results of the present study agree with those reported by Korchagin et al. (2020) [5] in vineyard and in forest soils, who observed that $Zn_R$ was the most abundant fraction and that $Zn_{OM}$ is the most variable fraction, with nutritional and environmental consequences. Interestingly, Vázquez-Blanco et al. (2022) [28] found opposite results for copper in the same vineyard sites, evidencing that the Cu concentrations declined after abandonment, and they tended to be associated with the less mobile fractions.

## 5. Conclusions

Zn is an essential nutrient that may be bounded to different geochemical fractions, and may provoke phytotoxicity depending on its mobility. The fractionation of Zn in soil may provide information about its potential transformation into soil. Land-use changes after vineyard abandonment into forests may vary Zn fractionation, resulting in a change from Zn bounded to the most immobile fractions in active vineyards towards an increase in Zn associated with SOM fractions in abandoned vineyards, especially in the uppermost soil layers. This study showed that a change in the land-use type resulted in an increase of nearly 25% in the geochemical mobility of Zn in abandoned vineyards compared to active ones. This means that a land-use change may trigger the potential toxicity of Zn in the soil environment, whose intensity would depend on the degree of SOM humification, stabilization, and mineralization. Therefore, caution must be taken on behalf of stakeholder authorities when abandonment occurs and/or when a land-use change is authorized.

**Supplementary Materials:** The following supporting information can be downloaded at https://www.mdpi.com/article/10.3390/horticulturae9101121/s1: Table S1: Soil chemical properties of all studied vineyards at 10 depths classified by site and land use.

**Author Contributions:** Conceptualization, P.P.-R., J.C.N.-M., M.A.-E. and D.F.-C.; methodology, P.P.-R., J.C.N.-M. and M.A.-E.; formal analysis, P.P.-R., J.C.N.-M. and D.F.-C.; investigation, P.P.-R. and J.C.N.-M.; resources, P.P.-R. and J.C.N.-M.; data curation, P.P.-R., J.C.N.-M., M.A.-E. and D.F.-C.; writing—original draft preparation, P.P.-R.; writing—review and editing, P.P.-R., J.C.N.-M., M.A.-E. and D.F.-C.; visualization, P.P.-R., J.C.N.-M., M.A.-E. and D.F.-C.; supervision, P.P.-R., J.C.N.-M., M.A.-E. and D.F.-C.; project administration, J.C.N.-M. and M.A.-E.; funding acquisition, J.C.N.-M. and M.A.-E. All authors have read and agreed to the published version of the manuscript.

**Funding:** This research was funded by Consellería de Cultura, Educación e Universidade (Xunta de Galicia), grant number ED431C2021/46-GRC. Paula Pérez-Rodríguez was funded by Juan de la Cierva contract, grant number JC2020-044426-I, from the Spanish Ministry of Science and Innovation and NextGenerationEU. The APC was partially funded by University of Vigo.

**Acknowledgments:** Paula Pérez-Rodríguez acknowledges her Juan de la Cierva contract from the Spanish Ministry of Science and Innovation and NextGenerationEU. The financial support of the Consellería de Cultura, Educación e Universidade (Xunta de Galicia), granted to the research group BV1 of the University of Vigo is also acknowledged.

**Conflicts of Interest:** The authors declare no conflict of interest.

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
