# Peer review of "Soil Abandonment as a Trigger for Changes in Zn Fractionation in Afforested Former Vineyard Acidic Soils"

_horticulturae, doi:10.3390/horticulturae9101121_

Round 1

Reviewer 1 Report

(1)   Why does the authors choose soil abandonment, considering for the complex composition of soil abandonment?

(2)   Many abbreviations require full names, please carefully verify by the author.

(3)   Lead is generally associated with zinc. Has the author considered the presence of lead and its impact on zinc?

(4)   There are no error bars in the figures, how can we ensure the credibility of the data?

(5)   The information provided by the image to the reader is not direct. It is recommended to enhance the expressive effect of the images.

 Minor editing of English language required

Author Response

Thank you for your insightful comments. We have addressed all to the best of our knowledge.

Reviewer 2 Report

You have got some surprising results. Whereever I analyzed Zn, it was one of the more mobile metal ions. Some questions and remarks:

Line 51: "available Zn"  .... please define! Is it exchangeable, or EDTA-Zn, or DCTA-Zn, or else?

Line 55: incorrect citations. Austria has not been mentioned in [20-22]. In Austria, the means soil contents for Zn (aqua regia, < 2mm) in regions suitable for vineyards, ranges within 56-76 mg/kg. See O.H. Danneberg, Mitt. d. Österr. Bodenkundl. Ges. 57, 7-24 (1999).

Citation [20] is Arch.Agron. Soil Sci. 2014, 60(issue 5), 609-624 .... is correct, Zn in natural field soil had been found at only 5 mg/kg , and in EDTA about 3 mg/kg

Citation [21] is J. Hazardous Mat. 2022, 424 ... is not correct. Total Zn was found ranging between 17 and 85 mg/kg, like in background of European soils. Zn extracted by DTPA-CaCl2 ("CAT") ranges 4-10 mg/kg

Citation [22] is Modeling Earth Systems and Environment 2022, 8, 407-416 .... is not correct. The given values for Zn-mean = 2,61 mg/kg (range 0,10-6,99) refer to the DTPA-Extraction, which will yield about 45 mg/kg, when the formula given in [21] gets applied.

Lines 164-165: give molar concentrations instead of normal 

Lines 198, 208, and 322: extraordinary high concentrations, especially for the residues! Within the moderately contaminated Danube sediments, I had found about  50 +/-13 % Zn-residual, in soils max. 40 %. My explanation is, this is immobilized fungicide Mancozeb (or another one was used?) - please check! Due to Wikipedia, the solubility  in water is just 6 μg/l, and after evaporation of the organic solvent, the Mancozeb (manganese- zinc-ethylenebis(dithiocarbamate) gets fixed. ZnS would be easily oxidized by O2, when it gets wet. But your methods do not target ZnS

Please check, if you can remove the residual Zn with organic solvents! Remove water from the soil-capillaries at first.

Mancozeb = Manganese-Zinc-Ethylenebis(dithiocarbamate)

ZnHPO 4 und Zn 3(PO4)2 would be soluble in oxalate pH3 

Author Response

(The authors gave the same response as above.)

Round 2

Reviewer 1 Report

The author has carefully revised and the current version can be accepted.